# Fibroblast Growth Factor 23 Signaling Does Not Increase Inflammation from *Pseudomonas aeruginosa* Infection in the Cystic Fibrosis Bronchial Epithelium

**DOI:** 10.3390/medicina59091635

**Published:** 2023-09-09

**Authors:** Meghan June Hirsch, Emma Lea Matthews, Seth Bollenbecker, Molly Easter, Megan R. Kiedrowski, Jarrod W. Barnes, Stefanie Krick

**Affiliations:** 1Division of Pulmonary, Allergy and Critical Care Medicine, Department of Medicine, The University of Alabama at Birmingham, Birmingham, AL 35233, USA; hirsch25@uab.edu (M.J.H.);; 2Gregory Fleming James Cystic Fibrosis Center, The University of Alabama at Birmingham, Birmingham, AL 35233, USA

**Keywords:** cystic fibrosis, bronchial inflammation, *Pseudomonas aeruginosa*, fibroblast growth factor 23

## Abstract

*Background and Objectives*: Chronic inflammation due to *Pseudomonas aeruginosa* (PA) infection in people with cystic fibrosis (CF) remains a concerning issue in the wake of modulator therapy initiation. Given the perpetuating cycle of colonization, infection, chronic inflammation, and recurrent injury to the lung, there are increases in the risk for mortality in the CF population. We have previously shown that fibroblast growth factor (FGF) 23 can exaggerate transforming growth factor (TGF) beta-mediated bronchial inflammation in CF. Our study aims to shed light on whether FGF23 signaling also plays a role in PA infection of the CF bronchial epithelium. *Materials and Methods*: CF bronchial epithelial cells were pretreated with FGF23 or inhibitors for FGF receptors (FGFR) and then infected with different PA isolates. After infection, immunoblot analyses were performed on these samples to assess the levels of phosphorylated phospholipase C gamma (PLCγ), total PLCγ, phosphorylated extracellular signal-regulated kinase (ERK), and total ERK. Additionally, the expression of FGFRs and interleukins at the transcript level (RT-qPCR), as well as production of interleukin (IL)-6 and IL-8 at the protein level (ELISA) were determined. *Results*: Although there were decreases in isoform-specific FGFRs with increases in interleukins at the mRNA level as well as phosphorylated PLCγ and the production of IL-8 protein with PA infection, treatment with FGF23 or FGFR blockade did not alter downstream targets such as IL-6 and IL-8. *Conclusions*: FGF23 signaling does not seem to modulate the PA-mediated inflammatory response of the CF bronchial epithelium.

## 1. Introduction

Cystic Fibrosis (CF) is one of the most common genetic disorders caused by mutations in the cystic fibrosis conductance regulator (CFTR) gene, which encodes for an apical epithelial chloride channel [1,2]. CFTR dysfunction leads to a multisystemic disease mainly affecting the ciliated epithelium of the sinuses, the tracheobronchial tree, and the gastrointestinal tract [1,2,3]. The majority of people with CF (pwCF) experience respiratory symptoms due to defective mucociliary clearance and a consecutive buildup of mucus, leading to an optimal environment for bacterial colonization and infections [1,2,4]. *Pseudomonas aeruginosa* (PA)*,* an opportunistic bacterium, typically colonizes the lungs of pwCF by the age of 24–35 years, which leads to CF exacerbations with lung injury and scarring [5,6,7,8]. There have been many advances in the treatment of CF, including the optimization of respiratory clearance therapies and inhaled antibiotics, which have led to increased life expectancy and improved quality of life [9,10]. The introduction of highly effective CFTR modulator therapies [11,12] has led to an even higher life expectancy, but initial studies have shown that PA is not eradicated with extended treatment [13]. Therefore, PA still poses a threat to these individuals given its virulence and adaptability—especially in an aging population of pwCF, who are colonized with PA and have chronic lung disease. Additionally, chronic infection with a mucoid PA phenotype is associated with multi-drug resistance, poor clinical outcomes, and an increased risk of mortality [1,4,14,15]. Infections with PA lead to increased inflammation, which perpetuates a cycle of lung damage, continued infection, and further inflammation.

Fibroblast growth factors (FGFs) are a family of growth factors that are pleiotropic and bind to the four different FGF receptors (FGFR) 1–4 with different affinities [16,17]. FGF23 has been well characterized as a prognostic marker in chronic kidney disease (CKD) due to its impact on phosphate homeostasis, in addition to exerting pro-inflammatory effects. Studies have shown that FGF23 can induce left ventricular hypertrophy, which is mediated by FGFR4 [16,18,19,20]. Our lab has also shown that FGF signaling increases inflammation in several lung diseases; for example, FGF23 levels are upregulated in CF, chronic obstructive pulmonary disease (COPD), and pulmonary fibrosis and exerts a pro-inflammatory effect (in combination with TGF-β and cigarette smoke) in the CF and COPD bronchial epithelium. In COPD, FGF23 exerts its effects via FGFR4 signaling, whereas FGFR1 signaling is involved in the CF-associated airway inflammation [21,22].

Given that PA infection leads to increases in inflammation—particularly in the bronchial epithelium [23]—and that FGF signaling has been shown to be pro-inflammatory in lung diseases including CF, we aimed to determine whether the FGF23/FGFR signaling pathway was utilized by the CF bronchial epithelium in the context of PA infection. Determining if this pathway is utilized by the epithelium will provide broader context for FGF signaling in the CF lung and provide potential novel evidence for therapeutic targets targeting the inflammation and lung function decline caused by this bacterium.

## 2. Materials and Methods

### 2.1. Cell Culture and Stimulation

The CFBE41o- (CFBE) and 16HBE14o- (16HBE) cell lines were a generous gift from Dr. Megan Kiedrowski. The CFBE41o- cell line is homozygous for the Δ508 CFTR mutation. The 16HBE14o- cell line contains two normal CFTR genes. All CFBE41o- and 16HBE14o- cells were cultured and maintained in Minimum Essential Media (MEM) (Gibco, Billings, MT, USA, 11095-072) with the addition of 10% fetal bovine serum (Atlas Biologicals, Fort Collins, CO, USA, FP-0500-A), 0.5% Pen-Strep (Corning, NY, USA, 30-002-CI), 1% L-glutamine (Corning, 25-005-CI), and 0.2% Plasmocin (Invivogen, ant-mpp) at 37 °C with 5% CO_2_. Cell lines were cultured as previously described [23]. Where indicated, human recombinant FGF23 (40 ng/mL, RnD Systems, Minneapolis, MN, USA, 2604FG), AZD4547 (10 μM, Selleckchem, Houston, TX, USA, S2801) for (FGFR) 1–3 inhibition, or BLU9931 (0.1 μM, Selleckchem, S7819) for FGFR4 inhibition were used as a pre-treatment for 1 h before infection with or without PA.

### 2.2. P. aeruginosa Strains and Infection

The PA strain PAO1, a generous gift from Dr. Megan Kiedrowski, and mucoid clinical isolate PAM57-15—a generous gift from Dr. Susan Birket—were used for infection studies. PA strains were cultured in 5 mL of Luria Bertani (LB) broth (Fisher Bioreagents, Pittsburgh, PA, USA, L-30831) overnight at 37 °C, 1 day prior to the assay. The PA infection protocol was modified from the static co-culture biofilm assay method [23,24]. PA overnight cultures were washed and standardized to an multiplicity of infection of 2 × 10^−3^, then inoculated apically onto the surface of CFBE or 16HBE cells, as previously described [23]. They were then incubated for 24 h prior to harvest for specific assays.

### 2.3. ELISA

After infection with PA for 24 h, interleukin (IL)-6 or IL-8 was quantified in the basolateral media, apical media, and cell lysate prepared for IL-6 or IL-8 Human Uncoated ELISA (Invitrogen, Waltham, MA, USA, 88-7066-88 and 88-8086-88 respectively). ELISAs were performed according to the manufacturer’s protocol. The human IL-6 ELISA had a sensitivity of 2–200 pg/mL. The human IL-8 ELISA had a sensitivity of 2–250 pg/mL.

### 2.4. RNA Purification and Quantitative Real Time PCR (qRT-PCR)

After infection with PA for 24 h, total RNA was isolated using a GeneJET RNA Purification Kit (Thermo Scientific, Waltham, MA, USA, K0732). RNA concentrations in each sample were assessed using a Nanodrop (Thermo Scientific) and cDNA was synthesized using a Maxima^TM^ H Minus cDNA Synthesis Master Mix with a dsDNase kit (Thermo Fischer, Waltham, MA, USA, K1652). RT-qPCR was performed on an Applied Biosystems StepOnePlus using the TaqMan primers Interleukin-8 (Hs00174103_m1, CXCL8), Interleukin-6 (Hs00174131_m1, IL-6), Interleukin 1-β (Hs01555410_m1, IL-1β), FGFR1 (Hs00241111_m1, FGFR1), FGFR2 (Hs01552918_m1, FGFR2), FGFR3 (Hs00179829_m1, FGFR3), FGFR4 (Hs01106910_g1, FGFR4), Transforming growth factor-beta (Hs00998133_m1, TGFB1), klotho (Hs00934627_m1, KL), and reference gene glyceraldehyde 3-phosphate dehydrogenase (GAPDH). Fold changes were calculated via the ΔΔCT method previously described [23].

### 2.5. Protein Immunoblotting

All protein lysates were obtained from CFBE41o- or 16HBE14o- cells using radio immunoprecipitation assay (RIPA) buffer (RPI, R26200-250.0) with phosphatase inhibitor, phosphatase inhibitor cocktail II (RPI, P52102-1), and the protease inhibitor Roche cOmplete^TM^ Protease Inhibitor Cocktail (Millipore Sigma, St. Louis, MO, USA, 11836153001). Protein concentrations were determined by Bradford Assay. Subsequently, 40 μg of protein was loaded for each well. Proteins were separated on 4–20% precast Ready Gels (Bio-Rad, Hercules, CA, USA) and transferred onto nitrocellulose (Amersham, Staffordshire, UK, 10600003) or polyvinylidene difluoride (PVDF) membranes (Thermo Scientific, 88518). PVDF membranes were activated for at least 5 min in methanol. For the loading control, total protein was assessed by staining one gel after protein separation using GelCode^TM^ Blue Stain Reagent (Thermo Scientific, 24590) for 1 h. The gel was washed with deionized water overnight and imaged using an Amersham Imager 600 system (GE). Membranes were blocked with 5% Bovine Serum Albumin (BSA; Fisher bioreagents, BP9706-100) in Tris-buffered saline (pH 7.4) with 0.05% Tween 20 (TBST) (Fisher bioreagents, C58H114O26) for 30 min and incubated overnight with the following primary antibodies: rabbit total and phospho-anti PLCγ along with rabbit total and phospho-anti ERK (Cell Signaling Technologies, 2822S, 8713S, 4695S, and 9101S, respectively). After 3 washes with TBST, membranes were incubated with goat-anti-rabbit peroxidase conjugated (Invitrogen, 31466) at 1:6000 in TBST for 1 h. Positive signals were visualized by chemiluminescence on an Amersham Imager 600 system (GE) or Gel Doc XR Gel Documentation System (Bio-Rad, Hercules, CA, USA). Images were acquired using Image Lab 6.1 Software (Bio-Rad). Densitometry was measured using ImageJ software 1.53a (National Institutes of Health, Bethesda, MD, USA). Densitometry was quantified by dividing P-PLCγ by total PLCγ or P-ERK by total ERK. These data were then normalized by total protein from the GelCode^TM^ Blue Stain Reagent image.

### 2.6. Statistics

Data were analyzed using GraphPad Prism 9 for Macintosh v.9.5.1 (GraphPad Software, Boston, MA, USA). One-way or Two-way ANOVAs were performed, followed by Kruskal–Wallis tests, Tukey’s multiple comparisons, or Dunn’s post-hoc test using a 95% confidence interval where indicated. All graphs represent between *n* = 3–5 independent experiments. Data are expressed as means ± standard error of mean (SEM). Differences between groups were considered statistically significant if *p* < 0.05.

## 3. Results

### 3.1. FGF23 Did Not Further Increase the Expression of Pro-Inflammatory Markers in the PA-Infected CF Bronchial Epithelium

To determine the role of FGF23 signaling on the PA-infected CF bronchial epithelium, we isolated RNA and performed RT-qPCR to determine fold-change levels between uninfected and infected CFBEs. PA infection caused a significant decrease in FGFR 1, 2, and 3 expression when compared to the control. However, FGF23 treatment did not affect FGFR levels (Figure 1A). The pro-inflammatory markers interleukin (IL)-8 and IL-6 were also significantly elevated with PA infection, as previously described in the literature [3,23,25]—this was not affected by FGF23 treatment (Figure 1B). Interestingly, PA infection caused a significant downregulation of TGFβ and α-klotho expression—a key CF modifier gene and FGF23 co-receptor, respectively—which were also not altered by FGF23 treatment (Figure 1C). These results were also recapitulated in 16HBEs and showed similar effects, with no differences with the addition of FGF23 treatment with or without PA infection (Appendix A). In summary, FGF23 does not alter the mRNA expression levels of FGFRs or proinflammatory markers in the PA-infected CF or WT bronchial epithelium.

### 3.2. FGF23 Did Not Alter PA-Induced Pro-Inflammatory Marker Production and Secretion in the CF Bronchial Epithelium

To determine whether FGF23 exacerbated PA-mediated pro-inflammatory cytokine secretion, we analyzed the basolateral media, cell lysate, and apical media from CFBEs treated with or without PAM57-15 and with or without FGF23. IL-8 was significantly increased in the basal media, cell lysate, and apical media in the PA-infected cells (Figure 2A). However, there was no further increase when FGF23 treatment was added. Even though IL-6 secretion was similarly affected, a significant decrease in IL-6 secretion in the apical media in PA-infected cells was observed when compared to the uninfected controls (Figure 2B). Additionally, we analyzed the effects of FGF23 treatment in 16HBEs with or without PA infection and found a slight increase in IL-8 and IL-6 in response to PA infection. This was visualized by significantly increased levels of IL-8 in the apical media only and IL-6 in the cell lysate (Appendix A). Similar to the CFBEs, however, there was no statistically significant difference in IL-8 or IL-6 production in the 16HBEs with the addition of FGF23 (Appendix A)—therefore suggesting that FGF23 does not exacerbate the production of IL-6 or IL-8 in the bronchial epithelium in the context of PA infection.

### 3.3. Phosphorylation of PLCγ and ERK Was Not Significantly Affected by FGF23 in PA-Infected CFBEs

To assess downstream signaling pathways, we analyzed phospho-PLCγ, PLCγ, phospho-ERK, and ERK expression by Western blot in control and PA-infected CFBEs as these proteins are associated with the FGF signaling pathway. Our results indicated that phospho-PLCγ was significantly upregulated when infected with the clinical isolate PAM57-15. In addition, there was a trending increase with the PA lab strain PAO1 (Figure 3A). Moreover, we found trending decreases in phosphorylated ERK levels with the lab strain PAO1 compared to a slight increase in phosphorylated ERK with the PAM57-15 clinical isolate (Figure 3A). This suggests that activated PLCγ was increased by PA infection in the CF bronchial epithelium.

Furthermore, we were interested to see whether FGF23 would further increase the phosphorylation of PLCγ or ERK. We anticipated seeing further exacerbation of activated PLCγ, given Figure 3A [23]. Our results showed that FGF23 did not alter the phosphorylation of PLCγ or ERK in PAM57-15-infected cells (Figure 3B,C).

### 3.4. Isoform-Specific FGFR Blockade Did Not Alter the Phosphorylation of PLCγ or ERK or Pro-Inflammatory Cytokine Expression in PA-Infected CFBEs

To determine if the inhibition of FGFRs 1−3 would impact inflammation in the context of PA infection in CFBEs, we started by pre-treating CFBEs with the clinically used FGFR 1−3 inhibitor, AZD4547, at 10 μM for 1 h prior to infection with PA. As shown before, PAM57-15 increased the phosphorylation of PLCγ in CFBEs, but there was no significant decrease following FGFR1-3 inhibition (Figure 4A,B). The PAO1 strain caused a significant decrease in ERK phosphorylation when compared to the uninfected control without any modulation by AZD4547 (Figure 4A,B). Furthermore, FGFR inhibition did not affect IL-8 secretion in PA-infected CFBEs (Figure 4C). This suggests that PA-mediated downstream signaling and IL-8 secretion are independent from FGF23/FGFR signaling.

### 3.5. Inhibition of FGFR4 Did Not Decrease the Phosphorylation of PLCγ or ERK or Pro-Inflammatory Cytokine Expression in PA-Infected CFBEs

Given the differences seen in phospho-PLCγ due to PA infection alone compared to the uninfected control (Figure 3A), we inhibited FGFR4 using BLU9931—an FGFR4 isoform specific inhibitor—at 0.1 μM prior to PA infection for 1 h [22]. We then assessed the phosphorylation of ERK and PLCγ along with secretion of the pro-inflammatory marker IL-8. Inhibition of FGFR4 led to no changes between levels of phosphorylated PLCγ and ERK in the PA-infected group or cells infected with PA and BLU9931 treatment (Figure 5A,B). Although there were significant increases in IL-8 in the basal media, cell lysate, and apical media with PA infection alone, there were no significant differences between the PA infection group and the FGFR4-inhibited + PA-infected group. This suggests that FGFR4 is not involved in the PA-mediated IL-8 increase in the CF bronchial epithelium.

## 4. Discussion

Our results demonstrate that PA infection of CFBE ALI cultures leads to a significant increase in phosphorylated PLCγ, along with increased levels of the pro-inflammatory cytokines IL-6 and IL-8. However, these markers are not altered by the addition of FGF23, nor by the inhibition of receptors targeting the FGF signaling pathway. This suggests that both FGF23 and FGF signaling do not modulate PA infection in the bronchial epithelium. Our data is nevertheless surprising given the significant increase in activated PLCγ and the heightened pro-inflammatory response seen with PA infection of CFBE, which is not affected by FGF23. There is literature to suggest that the p38 MAP kinase/Syk kinase pathway is involved in inducing inflammation in the context of PA [26], which is likely independent of FGF23/FGFR signaling.

Additionally, PA infection affects the mRNA levels of certain subtype-specific FGFRs—namely, FGFR1, 2 and 3, but not FGFR4—although FGFR inhibition did not affect PA-mediated inflammation. Our data indicates decreased phosphorylation of ERK with an increase in phosphorylated PLCγ, independent of FGFR4. Although our lab has shown that the inhibition of PLCγ decreases levels of IL-8 production [23], these data suggest that this is not FGF23/FGFR-mediated. Ultimately, it seems that infection with PA does not involve FGF23/FGFR signaling and independently activates PLCγ in the CF bronchial epithelium. Given these results, further investigation studying upstream pathways for PLCγ is warranted. This is crucial to developing our understanding, as uncovering the pathway in which this occurs could lead to the identification of potential therapeutic targets to mitigate the excessive inflammation caused by PA infection.

Given that our lab and others have shown that FGF23 circulating levels are upregulated in several chronic inflammatory lung diseases and chronic kidney disease, FGF23 has been known to be a commonly used pro-inflammatory and prognostic marker. Given its role as a prognostic marker and its lack of effect on PA, further correlation studies involving these markers in serum or other tissue types will need to be examined—specifically in inflammatory lung diseases and chronic PA infection.

Furthermore, our lab has found that FGF23 by itself does not frequently lead to inflammation in the healthy bronchial epithelium. However, when present with other stimuli such as elevated phosphate in the context of chronic kidney disease or alpha klotho in the context of idiopathic pulmonary fibrosis, FGF23 can exert a protective effect [21,22]. It is well known that FGF23 circulating levels are upregulated in chronic kidney disease, and we have shown recently that FGF23 per se does not exert a pro-inflammatory effect on the “healthy” bronchial epithelium, whereas the COPD epithelium shows increases in IL-1β when stimulated with FGF23. Interestingly though, elevations in phosphate—which are commonly seen in CKD during later stages—led to increases of IL-6 and IL-8 in the bronchial epithelium, which was attenuated by FGF23. In a similar beneficial role, circulating and pulmonary FGF23 levels are elevated in patients with idiopathic pulmonary fibrosis, but FGF23 itself did not cause the upregulation of fibrotic genes in pulmonary fibroblasts. On the contrary, when given together with its co-receptor alpha klotho, TGF-β-induced fibrotic gene expression was attenuated [21,22]. Our study here did not show any effect on PA infection. Given this, there is a possibility that in the context of CFTR modulator therapy, FGF23 could play a protective role against inflammation with subsequent PA infection. This role will need to be further examined in subsequent studies.

## 5. Conclusions

Our study shows that PA infection elicits a pro-inflammatory response in the CF bronchial epithelium, which is not altered by FGF23 or FGFR inhibition. This finding contrasts with previous studies investigating CF airway cultures and demonstrating that FGF23 can exaggerate the TGF-induced inflammatory response. Despite this being a “negative” study, it is of importance to further characterize FGF23 and its pleiotropic effects in different lung cells and different chronic lung diseases. Additionally, continuing to uncover the mechanisms by which PA affects CF and control airways is of importance, as this bacterium continues to affect people with CF, non-CF bronchiectasis, and many more in both acute and chronic manners.

## Figures and Tables

**Figure 1 medicina-59-01635-f001:**
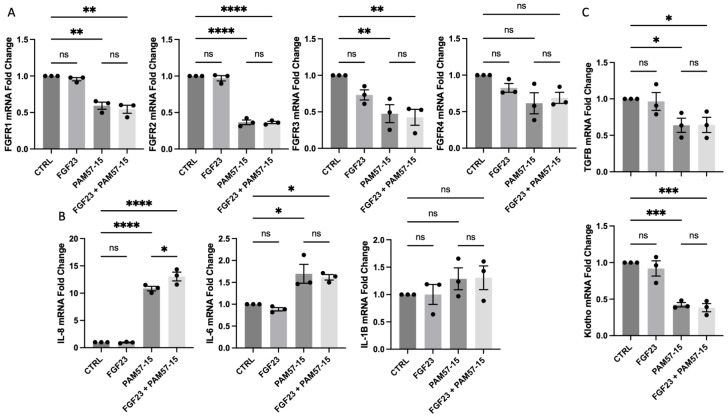
FGF23, in combination with PA infection, did not augment transcript expression of pro-inflammatory cytokines when compared to PA infection alone. (**A**) Fold change in mRNA levels of FGFRs 1–4 with or without FGF23 and PA (PAM57-15). (**B**) Fold change in mRNA levels of CXCL8 (IL-8), IL-6, and IL-1β after treatment with or without FGF23 and with or without PA. (**C**) Fold change in mRNA levels of TGFβ and klotho after treatment with or without FGF23 with or without PA infection. Data was represented as fold change in mRNA expression with *n* = 3 independent experiments. Statistical analysis was performed using a 2-way ANOVA, followed by Tukey’s multiple comparison post hoc test. Data are expressed as means ± standard error of the mean (SEM). Abbrv. ns = not significant. Differences were considered statistically significant if * *p* < 0.05, ** *p* < 0.01, *** *p* < 0.005, and **** *p* < 0.0001.

**Figure 2 medicina-59-01635-f002:**
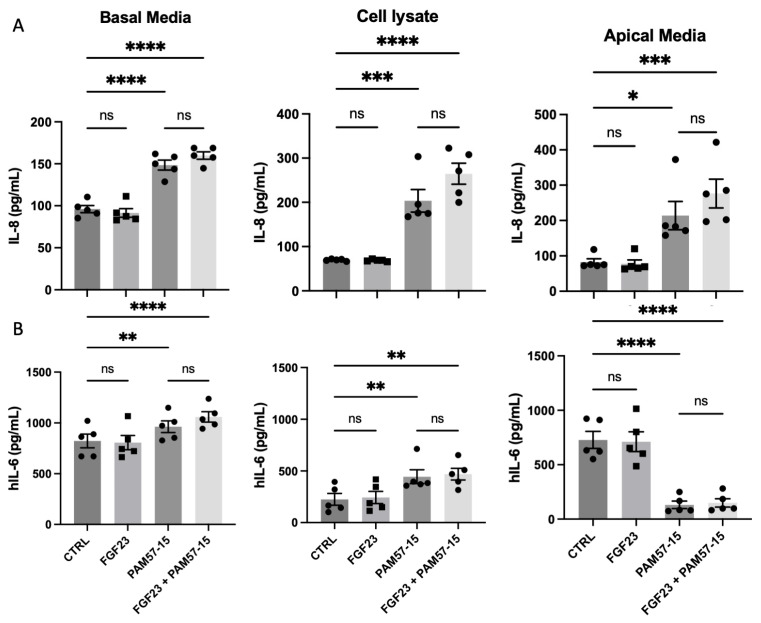
FGF23 did not alter pro-inflammatory marker production and secretion in PA-infected CFBEs. (**A**) Graph showing levels of IL-8 protein expression in the basolateral media, cell lysate, and apical media when CFBEs were treated with or without FGF23 and with or without PA infection. (**B**) Graphs showing levels of IL-6 protein expression in the basolateral media, cell lysate, and apical media when CFBEs were treated with or without FGF23 and with or without PA infection. Data was represented as expression (pg/mL) with *n* = 5 independent experiments. Statistical analysis was performed using a 2-way ANOVA, followed by Tukey’s multiple comparison post hoc test. Data are expressed as means ± standard error of the mean (SEM). Abbrv. ns = not significant. Differences were considered statistically significant if * *p* < 0.05, ** *p* < 0.01, *** *p* < 0.005, and **** *p* < 0.0001.

**Figure 3 medicina-59-01635-f003:**
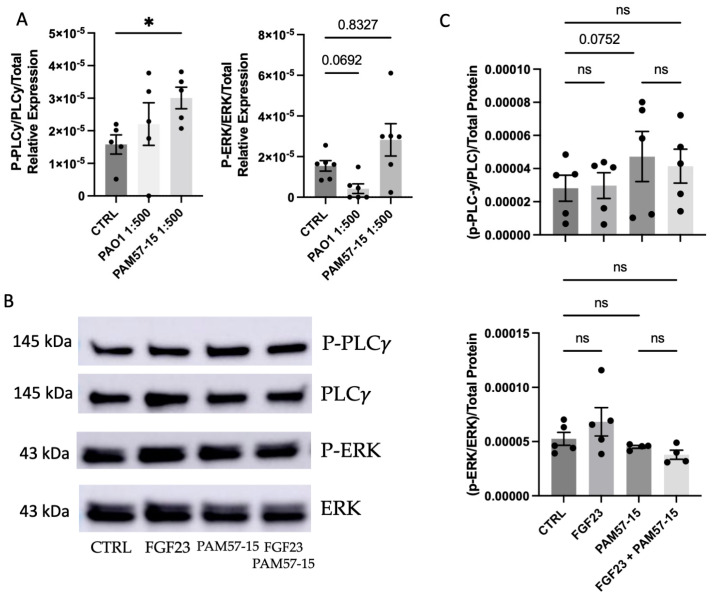
FGF23 did not alter the phosphorylation of PLCγ or ERK in the context of PA infection in CFBEs. (**A**) Quantification of Western Blots for phospho−ERK and phospho−PLCγ of PA-infected CFBEs compared to uninfected control CFBEs. (**B**) Representative western blots for phospho−PLCγ, total PLCγ, phospho−ERK, and total ERK with or without FGF23 treatment and with or without PA infection with the mucoid strain PAM57-15. (**C**) Densitometric quantification of Western Blot images from panel (**B**). Data was represented as phosphorylated PLCγ or ERK over their respective total, also normalized to total protein via the quantification of the Coomassie gel. These graphs represent *n* = 5−6 independent experiments. Statistical analysis was performed using a 1−way or 2−way ANOVA, followed by the Kruskal−Wallis test or Tukey’s multiple comparison post hoc test, respectively. Abbrv. ns = not significant. Data are expressed as means ± standard error of the mean (SEM). Differences were considered statistically significant if * *p* < 0.05.

**Figure 4 medicina-59-01635-f004:**
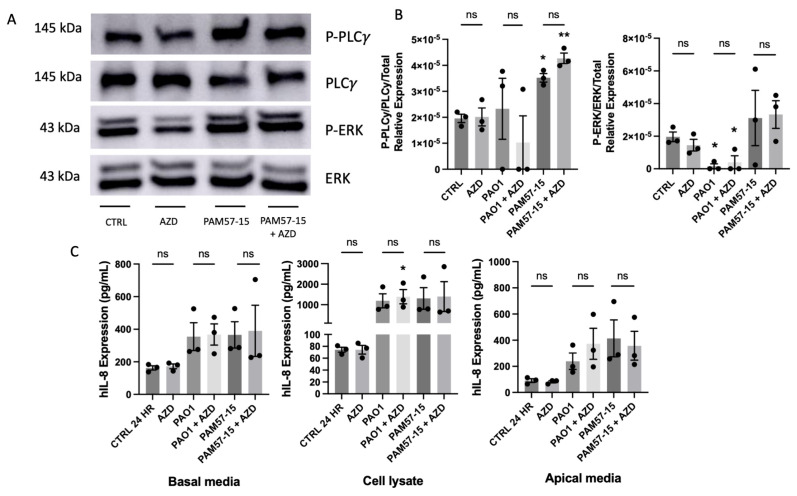
FGFR 1−3 inhibition did not decrease PLCγ or ERK phosphorylation or effect IL−8 production or secretion in PA−infected CFBEs. (**A**) Representative Western blot images of phospho−PLCγ, total PLCγ, phospho−ERK, and total ERK with or without FGFR inhibition (AZD) and with or without PA infection. (**B**) Densitometric quantification of Western blot images from panel (**A**). (**C**) IL−8 protein levels from basolateral media, cell lysate, and apical media of the CFBEs treated with or without FGFR 1−3 inhibition (AZD) and with or without PA infection. Western Blot data was represented as phosphorylated PLCγ or ERK over the respective total PLCγ or ERK, which was then normalized to total protein via the quantification of the Coomassie gel. These graphs represent *n* = 3 independent experiments. Statistical analysis was performed using a 2−way ANOVA, followed by Tukey’s multiple comparison post hoc test. Data are expressed as means ± standard error of the mean (SEM). Asterisks above columns represent significance compared to the uninfected control. Abbrv. ns = not significant. Differences were considered statistically significant if * *p* < 0.05 and ** *p* < 0.01.

**Figure 5 medicina-59-01635-f005:**
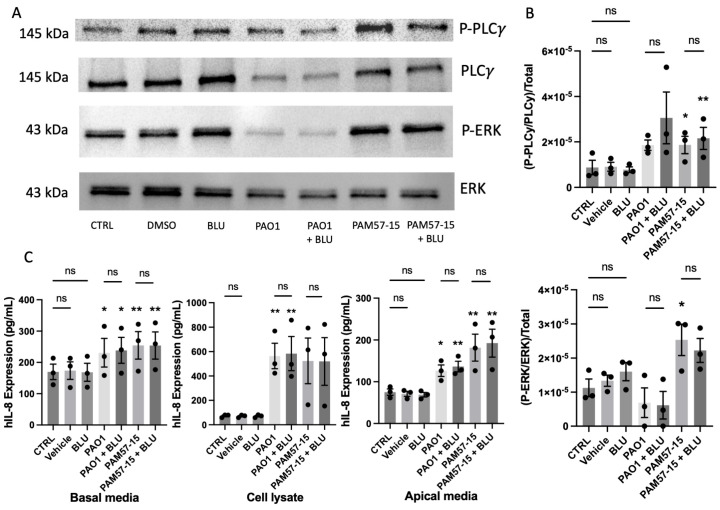
Inhibition of FGFR4 did not decrease the phosphorylation of PLCγ or ERK or decrease IL−8 production or secretion levels in PA−infected CFBE ALI cultures. (**A**) Representative Western blot images of phospho−PLCγ, total PLCγ, phospho−ERK, and total ERK with or without FGFR4 inhibitor and with or without PA infection. (**B**) Densitometric quantification of Western blots. (**C**) IL−8 protein levels from basolateral media, cell lysate, and apical media of CFBEs treated with or without R4 inhibition and with or without PA infection. Western Blot data was represented as phosphorylated PLCγ or ERK over the respective total PLCγ or ERK, which was then normalized to total protein via the quantification of the Coomassie gel. These graphs represent *n* = 3 independent experiments. Statistical analysis was performed using a 2−way ANOVA, followed by Tukey’s multiple comparison post hoc test. Data are expressed as means ± standard error of the mean (SEM). Asterisks above columns represent significance compared to the uninfected control. Abbrv. ns = not significant. Differences were considered statistically significant if * *p* < 0.05 and ** *p* < 0.01.

## Data Availability

The data generated or analyzed during this study are available from the corresponding author upon reasonable request.

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
