# Peer review of "Fibroblast Growth Factor 23 Signaling Does Not Increase Inflammation from Pseudomonas aeruginosa Infection in the Cystic Fibrosis Bronchial Epithelium"

_medicina, 2023, doi:10.3390/medicina59091635_

Round 1

Reviewer 1 Report

The authors aim to determine whether FGF23 signalling plays a role in PA infection of the CF bronchial epithelium. CF bronchial epithelial cells were pretreated with FGF23 or inhibitors for FGF receptors and then infected with different PA isolates. The authors found that here were increases in isoform specific FGFRs and interleukins at the mRNA level and phosphorylated PLC and production of IL-8 protein with PA infection, treatment with FGF23 or FGFR blockade did not alter downstream targets such as IL-6 and IL-8. 

This study carries an important value in better understanding the role of pro-inflammatory mediators in CF. However, please highlight the clinical application of this study, its limitations and future recommendations.

Similar studies have been widely carried out, and the authors need to highlight how this study adds to the current knowledge.

Reviewer 2 Report

The current manuscript from Dr. Stefanie Krick’s research group explored the role of Fibroblast Growth Factor (FGF) 23 signaling in Pseudomonas aeruginosa (PA) -mediated inflammation, using bronchial epithelial cells from cystic fibrosis (CF) patients. Authors concluded that FGF23 signaling is not involved in PA infection-mediated inflammation.

Although study observations are not indicating any involvement of FGF23 signaling, the experimental conditions can be further extended to explore the research question in more detail. But as a start of the study, the observations are important. Authors kept the research question simple and discrete. But future studies should consider more comprehensive experimental design.

Submission of the pictures of the full blots is greatly appreciated.

Suggestions

1.      An infection duration of 24 hours before subsequent treatment may be extreme. If the cells are infected for a shorter period, will there be any difference in the observations? Please comment.

2.      A dose response evaluation for both PA and FGF23 will be very useful to determine the most appropriate experimental condition.

3.      Did authors ever explore any effect of TRIKAFTA on the FGF23 signaling?

4.      Please include a “limitations” section in the discussion of the manuscript.

5.      Some of the datasets have very few technical replicates; if more numbers can be added, please consider presenting n=6 for all datasets.

6.      Please include protein size markers next to the blots presented in this manuscript.

7.      Figure 4A, please check the representative blots presented here.

8.      Please report the catalogue numbers of the reagents and kits used in the study.

Reviewer 3 Report

The sentences should be rewritten more more clearly

Round 2

Reviewer 3 Report

The authors have made necessary corrections and improved the quality of the paper.